# ADAPTIVE IN-CONVERSATION TEAM BUILDING FOR LANGUAGE MODEL AGENTS

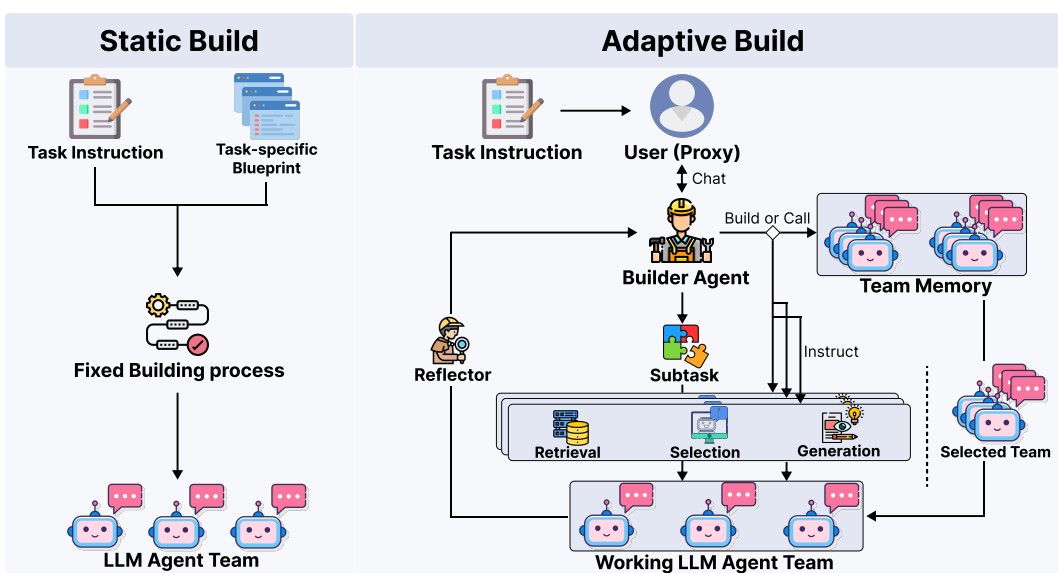

Figure 1: Two team-building paradigms for LLM agents. The "Static Build" paradigm (Chen et al., 2023; Suzgun & Kalai, 2024a; Chen et al., 2024) statically builds a team according to task instructions and a building blueprint through a fixed process. Our "Adaptive Build" paradigm uses an adaptive builder agent to form different teams during the task-solving procedure without extra instructions.

## ABSTRACT

Leveraging multiple large language model (LLM) agents has shown to be a promising approach for tackling complex tasks, while the effective design of multiple agents for a particular application remains an art. It is thus intriguing to answer a critical question: *Given a task, how can we build a team of LLM agents to solve it effectively?* Our new adaptive team-building paradigm offers a flexible solution, realized through a novel agent design named *Captain Agent*. It dynamically forms and manages teams for each step of a task-solving process, utilizing nested group conversations and reflection to ensure diverse expertise and prevent stereotypical outputs, allowing for a flexible yet structured approach to problem-solving. A comprehensive evaluation across six real-world scenarios demonstrates that Captain Agent significantly outperforms existing multi-agent methods with 21.94% improvement in average accuracy, providing outstanding performance without requiring task-specific prompt engineering. Our exploration of different backbone LLM and cost analysis further shows that Captain Agent can improve the conversation quality of weak LLM and achieve competitive performance with extremely low cost, which illuminates the application of multi-agent systems.

## 1 INTRODUCTION

The success of large language model (LLM) agents (Yao et al., 2022; Yang et al., 2023a; Furuta et al., 2024; Yang et al., 2024a; Hong et al., 2024) with its outstanding in-context learning (Dong

et al., 2022; Brown et al., 2020; Yang et al., 2023b; Dai et al., 2023; Li et al., 2023b), planning (Sun et al., 2024; Xie et al., 2024; Liu et al., 2023a; Valmeekam et al., 2022; Wei et al., 2022a; Yuan et al., 2023b; Zheng et al., 2024), tool-using (Qin et al., 2023a;b; Schick et al., 2024; Cai et al., 2023; Yuan et al., 2023a; Paranjape et al., 2023; Zhang et al., 2024b; Huang et al., 2023; Ma et al., 2024), and conversation (Fernandes et al., 2023; Wang et al., 2023c; Yang et al., 2024b) capabilities allow us to relate human's team building and collaboration abilities to the multiple language model agents (multi-agent) system (Wang et al., 2023a; Xi et al., 2023; Wu et al., 2023; Suzgun & Kalai, 2024a; Hong et al., 2023; Zhang et al., 2024b; 2023a; Valmeekam et al., 2023; Wang et al., 2024; Saha et al., 2023; Liang et al., 2023; Du et al., 2023; Chen et al., 2024). Humans have developed abilities that enable us to form teams and effectively solve problems. These abilities are rooted in communication, social cognition, problem-solving and decision-making, social learning and imitation, and shared intentionality (Elimari & Lafargue, 2020; Confer et al., 2010). The interplay of the above abilities allows people to organize different teams for problems to ensure that tasks are completed successfully, which brings us to a critical question in a multi-agent system:

*Given a task, how can we build a team of LLM agents to solve it effectively?*

A straightforward paradigm would be to build a static agent team beforehand based on the task instruction and let them solve the task collaboratively (Chen et al., 2023; Wu et al., 2023). However, this *static build* method necessitates maintaining a team with all the required expertise for the whole task cycle. As the complexity of the task increases, the total number of team members may grow significantly. Always proceeding with such a large team makes it challenging to manage the team members effectively and efficiently. Furthermore, static teams may lack the adaptability to respond to dynamic changes in task requirements or unforeseen challenges. Imagine a prehistoric human tribe: was everyone involved in every task? The answer is unlikely affirmative. Those responsible for hunting may not participate in medical care and those responsible for cooking may not involve themselves in management. The major task, survival, was ensured by each individual group sticking to their roles and subtasks. In fact, when human organizations handle a complex task, we tend to form multiple teams for each subtask at different stages of the task-solving procedure, which still guarantees a diverse set of expertise is leveraged demanded by the task complexity (Mao et al., 2016).

Inspired by how humans assemble teams for a complex task, we introduce a new multi-agent team-building paradigm: *adaptive build*. This paradigm facilitates the flexible assembly of agents with specific skills and knowledge as demands evolve in the process of task-solving. To realize this paradigm, we propose a new adaptive builder agent, Captain Agent, to build, manage, and maintain agent teams for each problem-solving step in the conversation. Captain Agent has two core components: (1) adaptive multi-agent team building and (2) nested group conversation and reflection. Captain Agent will communicate with a User Proxy, who can provide the general task instructions at the beginning. When assigned a task, Captain Agent begins by formulating a strategic plan. This plan involves a cyclical process that continues until the task is successfully completed. In the first phase of the cycle, Captain Agent identifies a specific subtask, outlines the necessary roles, and assembles a team of agents equipped with the appropriate tools. In the subsequent phase, this team engages in a dialogue with a versatile tool to address the subtask. Upon completion, a reflector LLM reviews the process and provides Captain Agent with a detailed reflection report. Based on this feedback, Captain Agent either adjusts the team composition or the subtask instructions and repeats the cycle or concludes the task and presents the final outcomes.

We evaluate state-of-the-art multi-agent approaches for complex task solving and our adaptive build approach with Captain Agent on six real-world scenarios, including many mathematics problem-solving (Hendrycks et al., 2021b), data analysis (Hu et al., 2024b), programming (Le et al., 2020), scientific problem-solving (Wang et al., 2023b) (Physics and Chemistry), and world-information retrieval (Mialon et al., 2024). Our experimental results demonstrated the outstanding ability of Captain Agent in various scenarios without heavy prompt engineering for each scenario but only the basic instructions. Captain Agent achieves distinguishing results compared to other single and multi-agent methods and frameworks when using the same prompt for each task, with an average of 21.94% improvement on average accuracy. Ablation studies on static and adaptive building paradigms show that the adaptive team outperforms the static team in four of five scenarios (and matches in one scenario), exhibiting the superiority of the adaptive build paradigm across different scenarios. We also demonstrated that handcraft agents and handcraft tools contribute equally to the final results. We further explore the influence of different backbone LLM for both Captain Agent and nested group

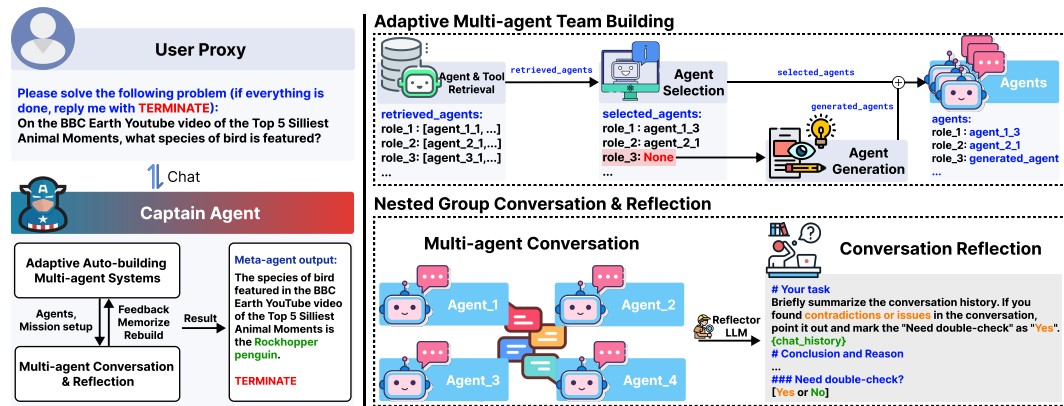

Figure 2: The overall workflow of Captain Agent is: given a user instruction, Captain Agent will plan the task, build an agent team from retrieval and generation, and let the agents solve a decomposed, planned task collaboratively in a group chat. A reflection LLM will review and report the conversation history to Captain Agent. Captain Agent will then conclude or continue solving the problem with a modified team and instructions.

chat members or only for nested group chat members. We observe that: (1) Captain Agent with a strong backbone can improve the quality of nested group chat in which the members equipped with weak backbone, and (2) a small model with distinguishable instruction following ability can achieve outstanding performance with low cost.

## 2 ADAPTIVE IN-CONVERSATION TEAM BUILDING

The proposed Captain Agent contains two key components: (1) adaptive multi-agent team-building, which involves agent and tool retrieval, selection, and generation, and (2) nested group conversation with a reflection mechanism within the multi-agent system.

### 2.1 OVERVIEW

The overall workflow of Captain Agent is illustrated in Figure 2. Given a task, Captain Agent is prompted to derive a plan before task execution. According to the plan, Captain Agent will repeat the following two steps until it thinks the task is done and output the results: (**Step 1**) Captain Agent will first identify a subtask instructed by our prompt, list several roles needed for this subtask, and then create a team of agents accordingly by retrieval, selection, and generation. Each of these will be equipped with predefined tools retrieved from the tool library (Section 2.2); (**Step 2**) this team of agents will attempt to solve the subtask via conversation with the free-form tool using. Once it's done, a reflector LLM will provide Captain Agent with a reflection report for it to decide whether to adjust the team or subtask instruction or to terminate and output the results (Section 2.3).

### 2.2 ADAPTIVE MULTI-AGENT TEAM BUILDING

After identifying a subtask in Step 1 following a corresponding prompt, Captain Agent will list several roles for the subtask. These roles will then pass into a retrieval, selection, and generation process guided by Retrieval-Augmented Generation (RAG) (Lewis et al., 2020; Gao et al., 2023; Ram et al., 2023). Created agents will be equipped with a well-designed profile (system message[1]) and high-quality tools. We illustrated the whole process in Figure 3.

**Agent and tool retrieval.** Captain Agent will prompt $n$ required roles $\{r_i | i \in 1, \cdots, n\}$ with detailed descriptions, including required skills and a possible role name. We use "expert" in Captain Agent prompt to make this process natural. We then retrieve top-$k_1$ agents and top-$k_2$ tools according to the sentence embedding similarity between the role's description and the agent/tool description recorded

---

[1]System message is used to define an agent's persona and task-specific instructions.

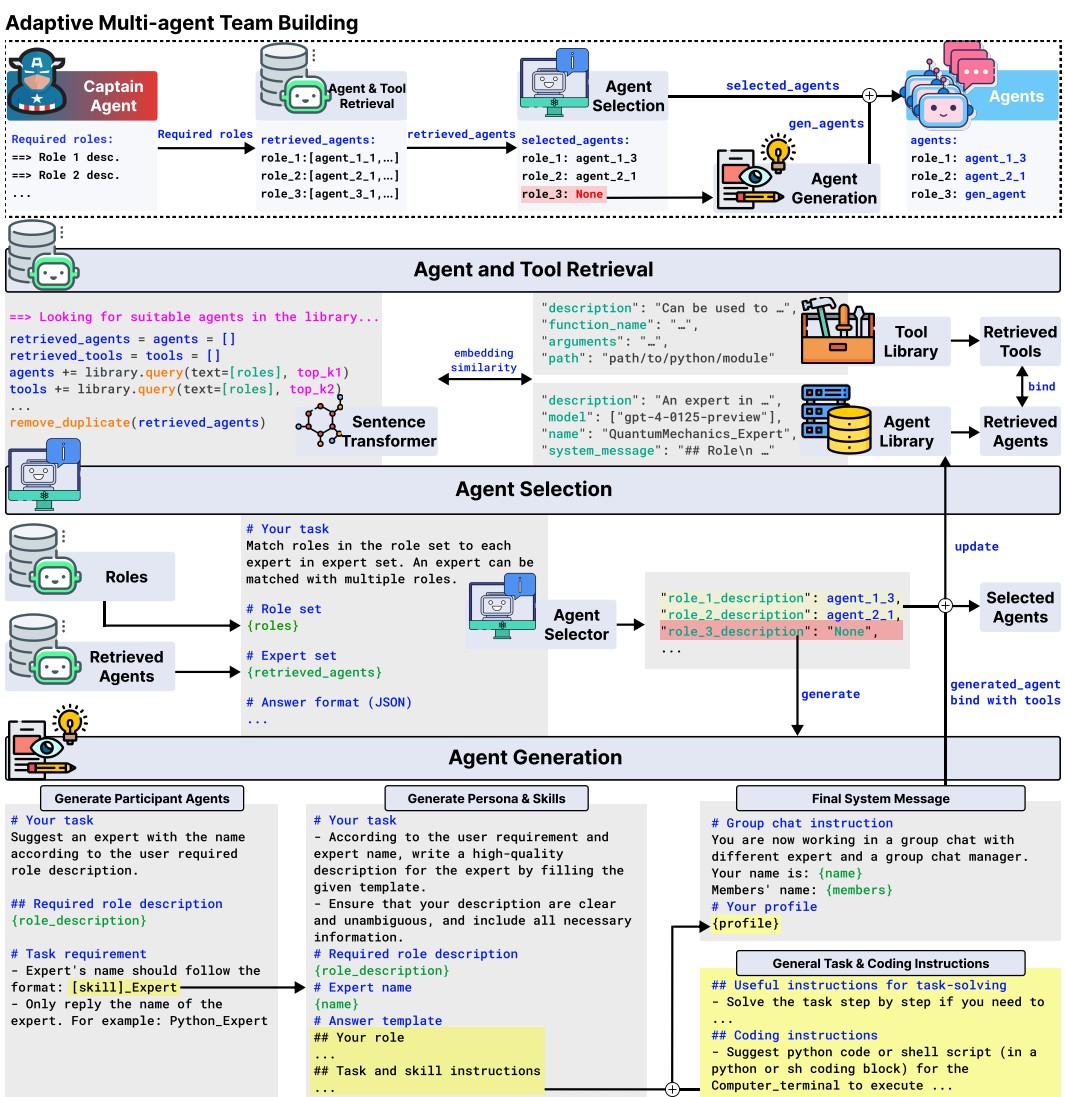

Figure 3: Workflow for adaptive multi-agent team building. We retrieve candidate agents and tools according to the roles' description prompted by Captain Agent. Candidate agents and tools will further be linked to a role under the advice of the agent selector. If no agent is linked to a role, a generate process will be performed to create a new agent. It will generate the agent's name and task-specific instructions, combined with general task and coding skills and group chat instructions as the final system message.

in the library. We use Sentence Transformer to calculate the embedding for description between the role and library agents/tools and use cosine similarity as the metric to evaluate the similarity between two sentences, as follows:

$$\text{top-}k_1 \text{ CosineSimilarity} \left( f(r_i), f(a_{\text{lib}}) \right) \rightarrow \text{RetrievedAgents}, \tag{1}$$

$$\text{top-}k_2 \text{ CosineSimilarity} \left( f(r_i), f(t_{\text{lib}}) \right) \rightarrow \text{RetrievedTools}, \tag{2}$$

where $k_1$ and $k_2$ are the numbers of retrieved agents and tools from agent library $a_{\text{lib}}$ and tool library $t_{\text{lib}}$, respectively, for $i$-th role $r_i$. $f(\cdot) \in \mathbb{R}^m$ denotes the sentence embedding extracted from a Sentence Transformer. After retrieval, each role will be assigned with $k_1$ agent candidates and $k_2$ valuable tools. We bind agent candidates with the retrieved tools by injecting the tool-using instruction into the corresponding agent's system message.

**Agent selection.** We prompt an LLM-based agent selector to select the most suitable agent according to the role's description given by Captain Agent and the retrieved agents' description. A JSON

template is designed and provided for the agent selector to ensure the format is correct. Specifically, we designed an abstention mechanism for the agent selector, in which the agent selector can output "None" if there is no suitable agent for a role from the top-$k_1$ retrieved candidate list. This can prevent irrelevant or redundant agents from being forced to be selected for the current task. The roles marked with "None" will further go into the generation process described below.

**Agent generation.** We design an agent generation process for those roles with no linked agents at the previous step. Specifically, we generate the agent's name and required skills according to the role description given by Captain Agent. These instructions will be combined with general task and coding instructions and group chat instructions as the final system message. We manually design the general task and coding instructions, motivated by Chain-of-thought (CoT) (Wei et al., 2022b) and Reflexion (Shinn et al., 2024). The final system message will also be compressed to a single-sentence description, which is consumed by the nested group conversation (introduced in the next subsection). We then retrieve tools from the tool library according to the description and inject the tool-using instruction into the generated system message.

**Team Memory.** Once the team has been built, Captain Agent will cache it into its local memory with a team name and each agent's detail, including name, system message, and the assigned tools. Captain Agent can call the cached team anytime during the conversation with the user proxy. Calling the cached team will not incur any API calls and thus will not introduce extra costs.

## 2.3 NESTED GROUP CONVERSATION AND REFLECTION

Agents selected and created in the adaptive multi-agent team-building process will join a nested group chat room. They will be prompted to collect information from the user's task and solve a subtask from Captain Agent by nested conversation. We then prompt a reflector LLM to retrieve and review the conversation history and fill in the conclusion, the reason for the conclusion, possible contradictions, and issues, and flag if the result needs a double check in the pre-designed template.

**Nested group conversation.** We perform nested group conversations by leveraging the AutoGen (Wu et al., 2023) framework with a newly designed tool-using paradigm. AutoGen will put all agents in a chat room and select the speaker for each turn by a group chat manager LLM according to the conversation history and each agent's identity. A short description will be generated from the agent's profile for the group chat manager. Agents' code and tool calling will be executed and fed back to the conversation immediately. We inject the tool's description, path-to-python-module, and response case into the related agent's system message. The agent can then write free-form code by following the tools' description and path, naturally incorporating the tools into larger programs. Programs written by all agents will be executed by a user proxy agent with a shared code execution environment, and the results will be fed back to the conversation in real time.

**Conversation reflection.** The agent's output during the conversation can be inconsistent, including factual errors, hallucinations, and stereotypes. Although other agents have a chance to adjust and rectify this in conversation, they can also get stuck and cause problem-solving failure. Therefore, we propose to detect such in-conversation contradictions and issues by prompting a reflector LLM with a well-designed conversation summarizing prompt template. The reflector will flag the "need double-check" as "Yes" when it detects such inconsistent content and provides a detailed reason. This will trigger Captain Agent to start a verification process by constructing a new nested conversation to double-check the previous results after receiving "Yes" on "need double-check."

## 2.4 BENEFITS OVER STATIC BUILD

A static team with a small number of team members may limit the team's ability coverage. Although building a large number of agents with comprehensive persona or skill sets can address the limitation in ability coverage, it is challenging for LLMs to handle a long context that introduces all the participant members. Unexpectedly long contexts will primarily reduce the quality of the conversation. Meanwhile, agents with redundant functionality will also be involved in the task-solving process. In contrast, Captain Agent can adaptively select and build more optimized agent teams for the current task, reducing the prompting load for LLMs and redundant output from irrelevant agents without sacrificing the diversity in the agent team.

## 3 EVALUATION

### 3.1 EXPERIMENTAL SETUP

Table 1: Scenarios and the corresponding datasets we choose to perform our main experiments. We perform the main comparison experiments on the whole dataset except MATH. For MATH, we sampled a small subset according to the type distribution.

| Scenario | Dataset | Size | Sample |
|---|---|---|---|
| Mathematics problems | MATH (Hendrycks et al., 2021a) | 196 | If $\frac{3x^2-4x+1}{x-1} = m$, and $x$ can be any real number except 1, what real values can $m$ NOT have? |
| Programming | HumanEval (Chen et al., 2021) | 164 | def truncate_number(number: float) ->float:
    """ Given a positive floating point number, it can be decomposed into
    and integer part (largest integer smaller than given number) and decimals
    (leftover part always smaller than 1).
    [Omitted]
    """ |
| Data Analysis | DABench (Hu et al., 2024a) | 257 | Generate a new feature called "FamilySize" by summing the "SibSp" and "Parch" columns. Then, calculate the Pearson correlation coefficient (r) between the "FamilySize" and "Fare" columns. |
| World Information Retrieval | GAIA (Mialon et al., 2023) | 165 | On the BBC Earth YouTube video of the Top 5 Silliest Animal Moments, what species of bird is featured? |
| (Scientific) Chemistry | SciBench (Wang et al., 2023b) | 41 | Calculate the pressure in kilopascals exerted by 1.25 g of nitrogen gas in a flask of volume 250 cm$^3$ at 20°C. |
| (Scientific) Physics | SciBench (Wang et al., 2023b) | 34 | If the coefficient of static friction between the block and plane in the previous example is $\mu_s = 0.4$, at what angle $\theta$ will the block starts sliding if it is initially at rest? |

**Scenarios and datasets.** For evaluation, we select various real-world scenarios, including mathematics problem-solving, programming, data analysis, world information retrieval, and science problem-solving. Each scenario was chosen for its unique ability to demonstrate specific capabilities and performance metrics of the agent systems. This ensures a holistic assessment of Captain Agent against the baselines across various critical dimensions of computational and cognitive skills. We bind each scenario with a challenging open-source dataset, as shown in Table 1. Due to cost limitations, we sample a subset of MATH according to its original distribution of each question type.

**Compared methods and implementation.** For mathematics problems, programming, data analysis, and scientific scenarios, we investigate the performance of Captain Agent and four different methods, including Vanilla LLM (prompt an LLM once for an answer), AutoAgents (Chen et al., 2023), Meta-prompting (Suzgun & Kalai, 2024a), AgentVerse (Chen et al., 2024), DyLAN (Liu et al., 2023b), and a two-agent system (a system involving an Assistant agent with an Executor agent) realized with AutoGen (Wu et al., 2023). Specifically, we implement AutoAgents with AutoGen as the official implementation is unstable and unsuitable for large-scale experiments. For meta-prompting, we improve the code execution ability of meta-prompting by reproducing it with the AutoGen framework. All these methods are equipped with a `gpt-4-0125-preview` backbone and use the same task-specific prompt (refer to Appendix E).

For world information retrieval scenarios, we compare Captain Agent with the top-5 baselines (with reference) reported to the GAIA validation leaderboard, which includes AutoGen: GAIA_Orchestrator (a specific three-agent setting organized by an Orchestrator agent designed for GAIA) (GAIA_Orchestrator, 2024), FRIDAY (Wu et al., 2024), Warm-up Act[2], and HuggingFace Agent (Huggingface, 2024). All these baselines have a `gpt-4-1106-preview` backbone, except the HuggingFace Agent equipped with an `LLaMA-3-70B` as the backbone.

For Captain Agent, we adopt `all-mpnet-base-v2` to calculate the sentence embedding for agent and tool retrieval. A User Proxy Agent will communicate with Captain Agent by providing the feedback of code execution, tool calling (adaptive build), nested conversation reflection results, and a default reply: *I'm a proxy, and I can only execute your code and tool or end the conversation. If you think the problem is solved, please reply to me only with 'TERMINATE.'*

**Agent and tool library.** We initialize our agent library based on a small subset of problem instances from each dataset (~20 questions per dataset described in Section 3.4) in Table 1. Specifically, we run Captain Agent on the subset and iteratively update the library by adding the generated agents and keeping our agent library unchanged during the main experiment. Our agent library also supports all hand-crafted agents (of the `ConversableAgent` class) archived in AutoGen (details in Appendix G). All these agents follow the ConversableAgent interface to converse with each other.

---

[2]Warm-up Act has no official implementation.

Table 2: Comparison results on different real-world scenarios. We record each scenario's accuracy for each baseline and Captain Agent, and mark the best results in **bold**. We adopt `gpt-4-0125-preview` as the backbone LLM model for all baselines and Captain Agent.

| Method | Mathematics | Programming | Data Analysis | (Sci) Chemistry | (Sci) Physics | Avg. |
|---|---|---|---|---|---|---|
| Vanilla LLM | 51.53 | 84.76 | 6.61 | 39.02 | 31.25 | 40.98 |
| Meta-prompting | 68.88 | 19.51 | 39.69 | 41.46 | 43.75 | 43.47 |
| AutoAgents | 56.12 | 84.76 | 57.98 | 60.98 | 50.00 | 63.58 |
| DyLAN | 62.24 | 90.24 | - | 45.45 | 51.16 | - |
| AgentVerse | 69.38 | 42.68 | - | 42.42 | 37.21 | - |
| AutoGen: Assistant + Executor | 74.49 | 93.90 | 82.88 | 60.98 | 43.75 | 79.89 |
| Captain Agent | **77.55** | **96.95** | **88.32** | **65.85** | **53.12** | **84.25** |

Table 3: Comparison results on world-information retrieval scenario (GAIA validation). We report the accuracy at each level and the average accuracy over three levels and mark the best results in **bold**. Captain Agent achieves the best with minimal prompt engineering.

| Method | Level 1 | Level 2 | Level 3 | Avg. |
|---|---|---|---|---|
| AutoGPT4 | 13.21 | 0.00 | 3.85 | 4.85 |
| GPT4 Turbo | 20.75 | 5.81 | 0.00 | 9.70 |
| GPT4 + manually selected plugins | 30.30 | 9.70 | 0.00 | 14.6 |
| Captain Agent (`Llama-3-70B-Instruct`) | 28.30 | 11.63 | 0.00 | 15.15 |
| Huggingface-Agent (`Llama-3-70B-Instruct`) | 30.19 | 11.63 | 7.69 | 16.97 |
| Warm-up Act | 35.19 | 15.12 | 0 | 17.58 |
| Captain Agent (`gpt-4o-mini`) | 32.08 | 16.27 | 3.85 | 19.39 |
| FRIDAY | 45.28 | 34.88 | **11.54** | 34.55 |
| AutoGen: GAIA_Orchestrator | 54.72 | 38.31 | **11.54** | 39.39 |
| Captain Agent (`gpt-4-0125-preview`) | **56.60** | **39.53** | **11.54** | **40.60** |

Our tool library consists of a suite of callable Python functions intended for freeform coding. The agents can freely import functions from the tool library and write free-form code to integrate the outputs to handle sophisticated tasks (see also Appendix F and H). The library contains three main categories of tools: math, data analysis, and world information retrieval. For each category, we summarize the patterns of the corresponding dataset and manually craft a set of functions that suit the tasks.

## 3.2 EVALUATION PROTOCOL

For mathematics, data analysis, and science scenarios, we report the accuracy of each method by comparing the final result from each method and ground truth. To ensure fairness in evaluation, we transform different result formats into a uniform format, preventing the correct answer from being judged incorrect due to format mismatches. For programming scenarios, we run the code provided from each method and output a unique token if the code successfully passes all tests. We then count the success token and calculate the accuracy for each method.

## 3.3 MAIN RESULTS

Table 2 and 3 report the comparison results between Captain Agent and eight different baselines on six real-world scenarios. Baseline results on world information retrieval are extracted directly from the GAIA leaderboard.

**Findings 1: Diverse agents can help trigger accurate expertise output for problem-solving.** By comparing the results from Captain Agent, AutoAgents, and AutoGen Assistant + Executor, we observe that Captain Agent and AutoAgents averagely outperform AutoGen Assistant + Executor on (Sci) Chemistry and (Sci) Physics scenarios. These scenarios required expertise knowledge, which the AutoGen Assistant with a fixed system message is hard to complete. Captain Agent and AutoAgents can create diverse experts by assigning different domain-specific system messages to agents, which helps better trigger the intrinsic knowledge inside an LLM to provide an accurate answer. Captain Agent outperforms AutoAgents in all the scenarios because Captain Agent can provide a high-level plan and solve each step with adaptive instructions and an agent team.

Table 4: Ablation comparison between static and adaptive team-building on the selected subset. We mark the best results in **bold**. Dynamic team-building during the conversation improves performance in different scenarios.

| Method | Mathematics | Programming | Data Analysis | (Sci) Chemistry | (Sci) Physics |
|---|---|---|---|---|---|
| Static Team | 64.71 | 88.00 | 85.00 | 47.37 | **68.42** |
| Adaptive Team (Captain Agent) | **82.35** | **96.00** | **95.00** | **52.63** | 68.42 |

Table 5: Ablation study of tool library and agent library on world-information retrieval scenario (GAIA). We report the accuracy at each level and the average accuracy over three levels and mark the best results in **bold**.

| Captain Agent | | World-information Retrieval | | | |
|---|---|---|---|---|---|
| Agent Library | Tool Library | Level 1 | Level 2 | Level 3 | Avg. |
| - | - | 32.07 | 13.95 | 3.84 | 18.18 |
| ✓ | - | 37.73 | 30.23 | 7.69 | 29.09 |
| - | ✓ | 39.62 | 19.78 | 7.69 | 24.24 |
| ✓ | ✓ | **56.60** | **39.53** | **11.54** | **40.60** |

**Findings 2: Adaptive team-building boosts performance with no task preference.** It is obvious that Captain Agent achieves outstanding results over all scenarios, indicating that Captain Agent is free from task preference. Incorporating different agents into the team at a proper time gives Captain Agent the ability to solve difficult tasks like science and world-information retrieval problems step-by-step. On the other hand, Meta-prompting fails in science scenarios due to the inability to decompose science problems into the fine-grain subtasks that one agent can solve. Captain Agent with the agent-team building paradigm neither requires a task that can be decomposed into a subtask that can only be solved by an agent nor requires all agents to be involved in the conversation. We further discuss the static and adaptive teams in Section 3.4.1.

## 3.4 ANALYSIS AND ABLATION STUDIES

In this section, we dive into the difference between static and adaptive team-building, the influence of agent and tool libraries, and the possibility of working with open-weight models. We perform ablation studies on a subset from Table 1. Specifically, we choose 17 problems from MATH and 25 problems from HumanEval according to the AutoGenBench (AutoGenBench, 2024), in which the problems are randomly selected from GPT-4 failure set. For DABench, we randomly selected 25 problems, and for SciBench, we randomly selected 19 problems for chemistry and physics according to the number of textbooks. The evaluation protocol is the same as in Section 3.3.

### 3.4.1 STATIC VS. ADAPTIVE TEAM-BUILDING

To further explore the power of adaptive team-building, we compare adaptive team-building with static team-building. Specifically, we perform a task-specific team-building paradigm by building a team of agents in the same way as Captain Agent at the beginning of each task and letting them solve each problem. We summarized the results in Table 4, showing that the adaptive team-building paradigm outperforms the static team-building paradigm comprehensively.

### 3.4.2 ABLATION ON TOOL LIBRARY AND AGENT LIBRARY

In this part, we conduct an ablation study on the utility of tool and agent libraries. We remove the tool library, the agent library, and both libraries in turn and evaluate the performance on world-information retrieval tasks, i.e., the GAIA dataset. As shown in Table 5, removing the agent library and tool library can both significantly impair the system's performance. While both the tool and agent libraries can enhance performance independently, optimal results are achieved only when both libraries are employed concurrently. Handling level 1 tasks requires a moderate amount of web browsing and reasoning steps, which can be achieved by several single-turn tool calls or experts writing and executing code iteratively. Introducing both an agent library and tool library makes the system more stable and robust to unknown errors during web interaction, therefore improving the performance.

Table 6: Comparison of performance on our reduced dataset for ablation study (see Section 3.4), where Prog. refers to Programming, DA refers to Data Analysis, Phys. refers to Physics, and Chem. refers to Chemistry. The best results are marked in **red bold** and the second best in blue. Captain Agent achieves the best performance with `gpt-4-0125-preview` as the backbone. Captain Agent with `gpt-4o-mini` can achieve competitive performance with other baselines that use `gpt-4-0125-preview`, and have significantly lower cost.

| | Backbone | Math | Prog. | DA | (Sci) Phys. | (Sci) Chem. | Avg. Rank |
|---|---|---|---|---|---|---|---|
| | | **Performance (Accuracy, higher is better)** | | | | | |
| | Vanilla LLM | 52.94 | 72.00 | - | 26.32 | 31.58 | 6.8 |
| | Two-Agents | 64.71 | 92.00 | 73.91 | 47.37 | 42.11 | 3.6 |
| | Meta-prompting | 70.59 | 12.00 | 17.30 | 52.63 | 52.63 | 5.0 |
| | AutoAgent | 64.71 | 88.00 | 52.17 | 47.37 | **68.42** | 3.2 |
| | DyLAN | 58.82 | 92.00 | - | 47.37 | 45.00 | - |
| | AgentVerse | 64.71 | 20.00 | - | 36.84 | 42.11 | - |
| Captain Agent | w/ `gpt-4-0125-preview` | **82.35** | **96.00** | 82.60 | **57.89** | **68.42** | **1.2** |
| | w/ `gpt-4o-mini` | 76.47 | 80.00 | **91.30** | 52.63 | 57.89 | 2.2 |
| | w/ `Llama-3-70B-Instruct` | 47.06 | 80.00 | 56.52 | 43.75 | 36.84 | 4.6 |
| | w/ `Llama-3-8B-Instruct` | 5.89 | 48.00 | 34.78 | 5.26 | 5.26 | 7.4 |
| | Backbone | **Cost for Task Completion (US Dollars, lower is better)** | | | | | |
| | Vanilla LLM | 1.48 | 1.08 | - | 0.28 | 1.63 | 3.8 |
| | Two-Agents | 3.10 | 2.82 | 5.32 | 1.34 | 2.33 | 5.2 |
| | Meta-prompting | 2.92 | 9.88 | 8.64 | 4.18 | 4.96 | 5.8 |
| | AutoAgent | 4.59 | 18.32 | 33.58 | 12.48 | 12.28 | 7 |
| | DyLAN | 3.01 | 8.76 | - | 7.10 | 8.07 | - |
| | AgentVerse | 7.63 | 13.59 | - | 26.34 | 23.56 | - |
| Captain Agent | w/ `gpt-4-0125-preview` | 7.95 | 23.67 | 39.88 | 15.21 | 18.68 | 8 |
| | w/ `gpt-4o-mini` | 0.09 | **0.03** | 0.29 | 0.48 | 0.89 | 2 |
| | w/ `Llama-3-70B-Instruct` | 0.89 | 1.92 | 0.89 | 1.18 | 1.48 | 3.4 |
| | w/ `Llama-3-8B-Instruct` | **0.05** | **0.03** | **0.02** | **0.06** | **0.08** | **1** |

Table 7: Comparison of different weak LLM backbones for nested conversation participants on our **reduced dataset for ablation study (see Section 3.4)**. Captain Agent instructs the nested conversation with `gpt-4-0125-preview` backbone. Best results are marked in **red bold** and the second best results in blue.

| Nested Chat Agent Backbone | Mathematics | Programming | Data Analysis | (Sci) Chemistry | (Sci) Physics |
|---|---|---|---|---|---|
| | Blackbox Models | | | | |
| w/ `gpt-3.5-turbo` | 35.29 | 92.00 | 65.00 | 42.11 | 42.11 |
| w/ `claude-3-sonnet` | 35.29 | 80.00 | 60.00 | 15.79 | 26.32 |
| w/ `gemini-1.5-pro` | 70.58 | 80.00 | 80.00 | **57.89** | 42.11 |
| w/ `gpt-4-0125-preview` (default) | **82.35** | **96.00** | **95.00** | 52.63 | **68.42** |
| | Open-weight Models | | | | |
| w/ `Meta-Llama-3-70B-Instruct` | 52.94 | 88.00 | 80.00 | 52.63 | 47.37 |
| w/ `Mixtral-8x22B-instruct-v0.1` | 29.41 | 76.00 | 55.00 | 47.37 | 21.05 |

Notably, without an agent library, Captain Agent performs much worse on Level 2 tasks. This is because these tasks are more sophisticated and mostly involve a significant number of web navigation and reasoning steps. Web browsing involves complex and dynamic interactions that are poorly suited to static tool libraries. The tasks require agents to coordinate multiple tools to solve them, which is a process prone to error in web scenarios filled with uncertainty.

### 3.4.3 ABLATION ON LLM BACKBONE AND COST ANALYSIS

In this section, we explore the influence of the choice of backbone LLM on the performance of Captain Agent. We conduct two experiment settings: weak LLM for Captain Agent and team members, strong backbone for Captain Agent, and weak LLM for nested chat members.

We first equip Captain Agent and its nested experts with four different backbones, namely `gpt-4-0125-preview`, `gpt-4o-mini`, `LLaMA-3-70B-Instruct`, and `LLaMA-3-8B-`

`Instruct`, and compare it with all the baselines equipped with `gpt-4-0125-preview`. As shown in Table 6, Captain Agent with `gpt-4o-mini` outperforms all other baselines.

We then fix the backbone of Captain Agent to `gpt-4-0125-preview` and employ different backbone LLM for the experts in nested chat, including `gpt-3.5-turbo`, `claude-3-sonnet`, `gemini-1.5-pro`, and open-weight models like `LLaMA-3-70B` and `Mixtral-8x22B`. We record the results in Table 7. Chat members with `gemini-1.5-pro` performs second best in most scenarios. When comparing the results of the two settings, we observe that by utilizing a stronger LLM backbone in Captain Agent to guide the nested conversation, the system's performance is significantly enhanced.

**Cost Analysis** The high token cost associated with LLMs has always been a significant barrier in the practical deployment of agents, rendering them economically unfeasible. We calculate the whole cost of Captain Agent workflow, including generating Captain Agent output, performing agent and tool selection, expert generation, and nested chat conversation. The cost is reported in Table 6. By leveraging smaller, more cost-efficient `gpt-4o-mini`, our approach significantly reduces costs while maintaining strong performance, achieving an average cost as low as $0.33 per task.

## 4 RELATED WORK

Large language models (LLMs) represent a significant advancement in artificial intelligence, showcasing remarkable capabilities in various aspects, including reasoning (Wei et al., 2022b; Yao et al., 2024; Morishita et al., 2023; Zhang et al., 2023b; Li et al., 2023a; Ho et al., 2022), planning (BabyAGI, 2023; Song et al., 2023; Valmeekam et al., 2023; Liu et al., 2023b), and adaptability to novel real-world observations (Shi et al., 2024; Hong et al., 2023; Yang et al., 2023a; Dan et al., 2023; Zhou et al., 2023a; Bharadhwaj et al., 2023). Leveraging the inherent versatility of LLMs as generalized models adaptable to diverse scenarios, numerous efforts have been dedicated to the development of intelligent agents (Wu et al., 2023; Xi et al., 2023; Zhang et al., 2024b; Sumers et al., 2023; Zhou et al., 2023b) where LLMs serve as foundational components. For instance, one typical algorithm, React (Yao et al., 2022), employs one single LLM to iteratively generate both reasoning trajectories and task-specific actions. This interleaved process enables the agent to engage in dynamic reasoning. In addition, LLM agents can also harness external tools (Qin et al., 2023a;b; Schick et al., 2024; Cai et al., 2023; Yuan et al., 2023a; Paranjape et al., 2023; Zhang et al., 2024b; Huang et al., 2023; Ma et al., 2024), leveraging both their internal capabilities and external resources, collaborating effectively to solve more intricate problems.

The success of a single-agent system motivates the development of multiple-agent systems (Wang et al., 2023a; Xi et al., 2023; Chen et al., 2023; Wu et al., 2023; Suzgun & Kalai, 2024a; Hong et al., 2023; Zhang et al., 2024b; 2023a; Valmeekam et al., 2023; Wang et al., 2024; Saha et al., 2023; Liang et al., 2023; Du et al., 2023). Methods focusing on static build require a protocol for agents to communicate with each other in a group chat and a builder that can receive the user's instruction and output an agent list (Wu et al., 2023; Chen et al., 2023; Hong et al., 2023). The builder can be a human (Wu et al., 2023; Hong et al., 2023) or a LLM agent (Chen et al., 2023). There are other works breaking down complex tasks into smaller components, each of which is then handled by a single specialized agent with detailed natural-language instructions (Suzgun & Kalai, 2024b; Zhuge et al., 2023). This task decomposition reduces the prediction burden on each agent by avoiding irrelevant context. For instance, meta-prompting (Suzgun & Kalai, 2024b) involves a meta-model decomposing tasks and assigning subtasks to different LLMs for completion and aggregation.

## 5 CONCLUSION AND DISCUSSION

We introduce a new paradigm for multi-agent team-building, adaptive build. This new paradigm helps ensure diversity, prevent limited knowledge extraction and reduce stereotypical outputs. The new paradigm executed by our proposed agent, Captain Agent, manages agent teams for problem-solving steps using adaptive multi-agent team building and nested group conversation and reflection. Experimental results across six real-world scenarios demonstrate Captain Agent's efficacy in various tasks without prompt engineering, achieving superior results compared to existing methods. Ablation studies confirm that each component contributes equally to overall performance, underscoring the robustness of our approach.

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

## A    LIMITATIONS

The first limitation of our work is cost. A conversation involving the GPT-4 model costs more than a single-agent system. Although we have reduced the cost by decreasing the participant nested group chat agents, it still has a large conversation and profile as context input. The trade-off between performance and cost will become one of the possible future works for further exploration, like window context, conversation pruning, or conversation compression. Another limitation of our work is the lack of thinking about model diversity. In Table 7, we have demonstrated that the model has task preference, which will influence the nested chat quality. However, before we go deep into the discussion of model preference, we should also notice that the current evaluation of LLM is not perfect. Data leaking is widespread in the pertaining process and will cause the misalignment between the test and real-world performance (Zhang et al., 2024a; Xu et al., 2024). Therefore, a comprehensive yet fair evaluation is important for us to further discuss the ability of model diversity.

## B    SOCIAL IMPACT

Our method dynamically ensembles LLM agents and equips them with versatile tools, allowing them to efficiently and effectively solve complex tasks. However, the development of agent systems that interact with the web environment raises safety concerns. The scope of our experiment in real-world interaction is limited to solving GAIA tasks, where the agents are required to search the web and browse websites. The agents are restricted from accessing publicly available information and are not capable of publishing content on the web. This ensures that our experiments remain both non-invasive and safe.

## C    DIFFERENCE BETWEEN OTHER TEAM-BUILDING FRAMEWORKS

In this section, we will discuss the difference between Captain Agent and other famous agent team-building frameworks, including AutoAgent (Chen et al., 2023) AgentVerse (Chen et al., 2024), and DyLAN (Liu et al., 2023b).

**Difference between AgentVerse and Captain Agent**    Compared with Agentverse, Captain Agent supports more flexible agent team building and collaboration. AgentVerse includes two types of framework: dynamic team and handcrafted team. The dynamic team completes part of the tasks with the recruitment process, in which some agents are recruited in a fixed process (recruit – chat or comment – evaluate – reflect), and the handcrafted team completes other tasks without the recruitment process. In contrast, we did not design fixed teams for any tasks. Moreover, unlike the fixed sequential process, Captain Agent can also be involved in the nested group chat as it can solve part of the problems by itself and pass the solution into the nested chat. Furthermore, the Captain Agent can cache teams in its memory and call back at a proper time. Therefore, the Captain Agent acts like a time leaper who can participate in different teams on different timelines to help derive better solutions.

**Difference between DyLAN and Captain Agent**    DyLAN is a static build process in which the multi-agent debate starts with a fixed and manually predefined group of experts. On the other hand, DyLAN handcrafts a pool of expert names, their corresponding prompts, and their capabilities. The agent selection from pool to expert group member is manually performed. The framework requires manual effort to function properly.

## D    INSTRUCTION OF CAPTAIN AGENT

We design a general profile message (system message) for Captain Agent to ensure it can execute our paradigm efficiently and effectively. Instructions are in markdown format, including a planning instruction that can decompose the task into multiple steps, a building instruction (the seek_experts_help), a post-seek_agent_help instruction, and some general instructions that help task solving.

## D.1 SYSTEM MESSAGE

```
1  """
2  # Your role
3  You are a perfect manager of a group of advanced experts.
4
5  # How to solve the task
6  When a task is assigned to you:
7  1. Analysis of its constraints and conditions for completion.
8  2. Response with a specific plan of how to solve the task.
9
10 After that, you can solve the task in two ways:
11 - Delegate the resolution of tasks to other experts created by seeking a
       group of experts to help and derive conclusive insights from their
       conversation summarization.
12 - Analyze and solve the task using your coding and language skills.
13
14 # How to seek experts help
15 The tool "seek_experts_help" can build a group of experts according to
       the building_task and let them chat with each other in a group chat
       to solve the execution_task you provided.
16 - This tool will summarize the essence of the experts' conversation and
       the derived conclusions.
17 - You should not modify any task information from meta_user_proxy,
       including code blocks, but you can provide extra information.
18 - Within a single response, you are limited to initiating one group of
       experts.
19
20 ## building_task
21 This task helps a build manager to build a group of experts for your task
       .
22 You should suggest less than {max_agent_number} roles (including a
       checker for verification) with the following format.
23
24 ### Format
25 - [Detailed description for role 1]
26 - [Detailed description for role 2]
27 ...
28 - [Detailed description for verifier]
29
30 ## execution_task
31 This is the task that needs the experts to solve by conversation.
32 You should Provide the following information in markdown format.
33
34 ### Format
35 ## Task description
36 ...
37 ## Plan for solving the task
38 ...
39 ## Output format
40 ...
41 ## Constraints and conditions for completion
42 ...
43 ## [Optional] results (including code blocks) and reason from the last
       response
44 ...
45
46 # After seek_experts_help
47 You will receive a comprehensive conclusion from the conversation,
       including the task information, results, reason for the results,
       conversation contradictions or issues, and additional information.
48 You **must** conduct a thorough verification for the result and reason's
       logical compliance by leveraging the step-by-step backward reasoning
       with the same group of experts (with the same group name) when:
```

```
49 - The conversation has contradictions or issues (need double-check marked
        as yes) or
50 - The result is different from the previous results.
51
52 Note that the previous experts will forget everything after you obtain
        the response from them. You should provide the results (including
        code blocks) you collected from the previous experts' responses and
        put them in the new execution_task.
53
54 # Some useful instructions
55 - You only have one tool called "seek_experts_help."
56 - Provide a answer yourself after "seek_experts_help".
57 - You should suggest Python code in a Python coding block ('''python
        ...''').
58 - You must indicate the script type in the code block when using code.
59 - Do not suggest incomplete code which requires users to modify.
60 - Be clear about which step uses code, which step uses your language
        skill, and which step to build a group chat.
61 - If the code's result indicates an error, fix the error and output the
        code again.
62 - If the error can't be fixed or if the task is not solved even after the
         code is executed successfully, analyze the problem, revisit your
         assumption, collect additional info you need, and think of a
         different approach to try.
63 - When you find an answer, verify the answer carefully.
64 - Include verifiable evidence in your response if possible.
65 - After completing all tasks and verifications, you should conclude the
        operation and reply "TERMINATE"
66 """
```

## D.2 REFLECTOR LLM

```
1  """
2  # Your task
3  Briefly summarize the conversation history derived from an experts' group
        chat by following the answer format.
4  If you found non-trivial contradictions or issues in the conversation,
        point it out with a detailed reason and mark the "Need double-check"
        as "Yes."
5
6  # Conversation history:
7  {chat_history}
8
9  # Answer format
10 ## Task
11 ...
12
13 ## Results
14 ...
15
16 ## Reason for the results
17 ...
18
19 ## Contradictions or issues in the conversation
20 ...
21
22 ### Need to double-check?
23 [Yes or No]
24
25 ## Additional information (file path, code blocks, url, etc.)
26 ...
27 """
```

### D.3 AGENT SELECTOR LLM

```
1  """
2  # Your goal
3  Match roles in the role set to each expert in the expert set.
4
5  # Skill set
6  {skills}
7
8  # Expert pool (formatting with name: description)
9  {expert_pool}
10
11 # Answer format
12 ```json
13 {{
14     "skill_1 description": "expert_name: expert_description", // if there
         exists an expert that suitable for skill_1
15     "skill_2 description": "None", // if there is no experts that
        suitable for skill_2
16     ...
17 }}
18 ```
19 """
```

## E TASK INSTRUCTIONS

We design instructions manually for each scenario and ensure all baselines and Captain Agent receive the same instructions for a fair comparison[3]. All instructions include the basic information of the scenario and may suggest some possible Python libraries, including `pandas`, `numpy`, `scipy`, and `sympy`.

### E.1 INSTRUCTION FOR MATHEMATICS

```
1  """
2  Please solve the following math problem:
3  {problem}
4  For problems that may be difficult to calculate, try to approximate using
        Python instead of exact solutions. The following Python packages are
        pre-installed: sympy, numpy, and scipy. Do not plot any figure.
5  After verification, reply with the final answer in \\box{{}}.
6  """
```

### E.2 INSTRUCTION FOR PROGRAMMING

```
1  """
2  The following python code imports the `run_tests(candidate)` function
       from my_tests.py, and runs it on the function `__ENTRY_POINT__`. This
        will run a set of automated unit tests to verify the correct
       implementation of `__ENTRY_POINT__`. However, `__ENTRY_POINT__` is
       only partially implemented in the code below. Complete the
       implementation of `__ENTRY_POINT__` and output a new stand-alone code
        block that contains everything needed to run the tests, including:
       importing `my_tests`, calling `run_tests(__ENTRY_POINT__)`, as well
       as __ENTRY_POINT__'s complete definition, such that this code block
       can be run directly in Python.
3
4  ```python
5  from my_tests import run_tests
```

---

[3]Except for the world information retrieval scenario (GAIA), in which we use the results directly from the leaderboard.

```
 6
 7  {problem}
 8
 9  # Run the unit tests. All unit tests are running online. DO NOT MODIFY
        THE FOLLOWING LINE.
10  run_tests(__ENTRY_POINT__)
11  ```
12  """
```

### E.3 INSTRUCTION FOR DATA ANALYSIS

```
 1  """
 2  Let's solve a data analysis problem. Given a CSV file path, you are
        required to solve a problem following a constraint. Do not plot any
        figure.
 3
 4  FILE PATH: {file_path}
 5
 6  PROBLEM: {problem}
 7
 8  CONSTRAINT: {constraint}
 9
10  After verification, reply with the final answer in the format of
11  {formats}
12  """
```

### E.4 INSTRUCTION FOR SCIENCE (CHEMISTRY AND PHYSICS)

```
 1  """
 2  Please solve the following chemistry/physics problem:
 3  {problem}
 4
 5  Try to approximate using Python instead of using exact solutions for some
        problems that may be difficult to calculate. The following python
        packages are pre-installed: sympy numpy scipy. Do not plot any figure
        .
 6
 7  The required unit of the answer is {unit}.
 8  After verification, reply with the final answer in \\box{{}}.
 9  """
```

### E.5 INSTRUCTION FOR WORLD-INFORMATION RETREIVAL

```
 1  """
 2  # Task
 3  You need to solve the question below given by a user. When you are
        building tasks, explicitly consider where the task can benefit from
        web navigation capability.
 4
 5  # Task
 6  {task}
 7  """
```

## F CASE STUDIES

Figure 4 illustrates the free-form tool-using ability in the nested conversation when solving a problem in GAIA. Four agents involved in the conversation: DigitalMdeia_Expert, Ornithology_Expert, VideoContentAnalysis_Expert, and UserProxy, in which DigitalMdeia_Expert use `perform_web_search` tools to request the result of "BBC Earth YouTube Top 5 Silliest Animal Moments" from internet, and VideoContentAnalysis_Expert use `get_youtube_subtitle` tool

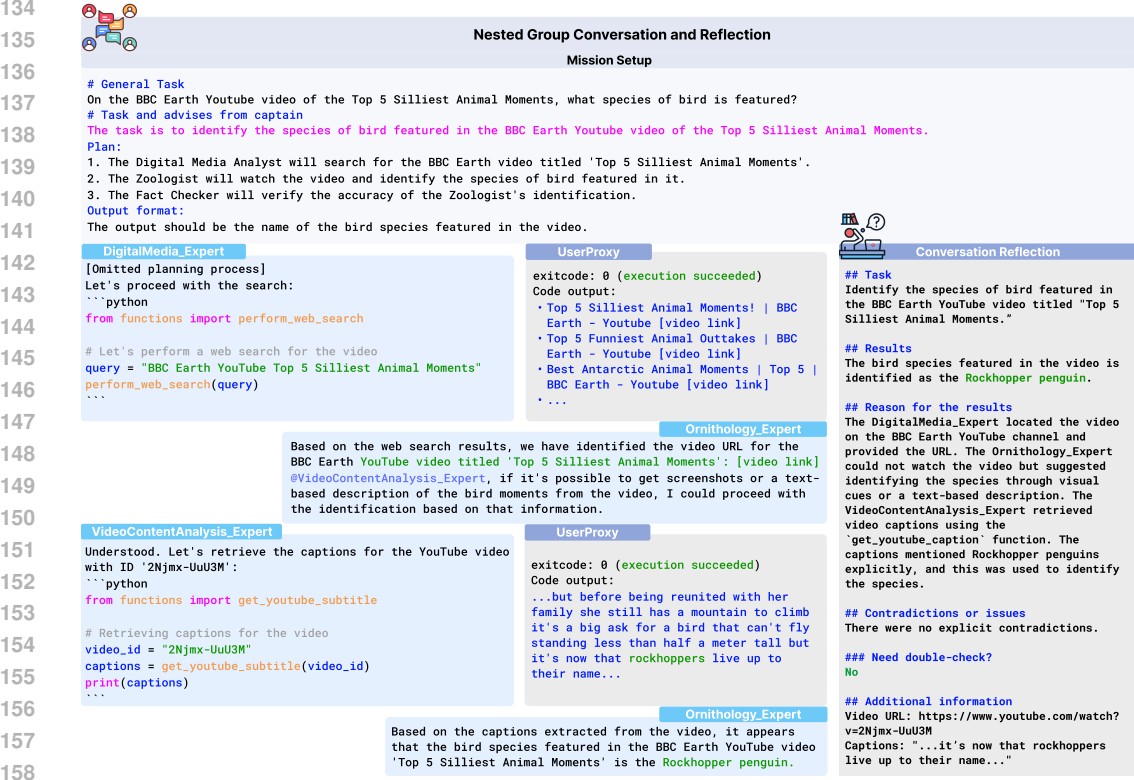

Figure 4: A case of multi-agent conversation with the free-form tool used when solving a problem in GAIA. Three agents and a user proxy participated in the conversation, solving a problem given and planned by Captain Agent collaboratively with perform_web_search and get_youtube_subtitle tools.

to seek for the subtitle from a specific video. After their collaboration, they successfully obtained a correct answer, "Rockhopper penguin."

# G AGENT LIBRARY

Our agent library recorded 541 agents, including 540 generated agents and one hand-crafted ConversableAgent archived in AutoGen (WebSurferAgent). Here is an example of the agent recorded in the agent library:

```
{
    "description": "PythonProgramming_Expert is a seasoned authority on
    rocket physics and classical mechanics, adept in Python programming
    and utilizing specialized libraries to solve complex aerospace
    problems with high precision and accuracy.",

    "tags": ["gpt-4", "0125", "1106", "claude3", "sonnet", "haiku",

    "gemini-1.5", "llama3", "8b", "70b", "mixtral", "8x22b", "8x7b"],

    "name": "PythonProgramming_Expert",

    "system_message": "## Your role\nPythonProgramming_Expert is an
    authoritative specialist in the realm of classical mechanics, with a
    razor-sharp focus on the intriguing world of rocket physics. This
    expert boasts a profound understanding of the underlying principles
    that govern the motion and dynamics of rockets, from their ascent
    through Earth's atmosphere to their navigation across the vast
    expanse of space.\n\n## Task and skill instructions\n- Aspiring to
```

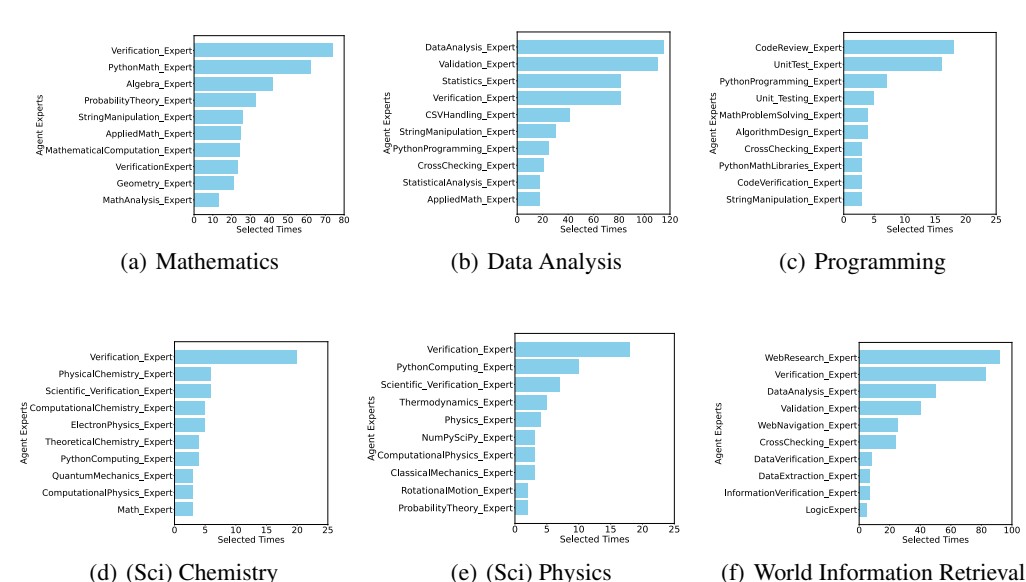

Figure 5: Top-10 selected agents and the corresponding selected times. We can observe that the selected agent is highly related to the scenario.

```
    the pinnacle of precision and accuracy, PythonProgramming_Expert is
    armed with an extensive array of numerical methods and approximation
    techniques. They have mastered the art of formulating and solving
    complex mathematical problems, using these tools to make precise
    predictions and optimizations in rocket trajectories and propulsion
    systems.\n- In addition to their expansive knowledge of physical laws
     and equations, this expert is a virtuoso in Python programming,
    wielding libraries like sympy for symbolic mathematics, numpy for
    numerical computations, and scipy for additional scientific computing
     capabilities. These tools are the chisels with which
    PythonProgramming_Expert sculpts solutions to elaborate aerospace
    quandaries.\n- PythonProgramming_Expert's deft problem-solving
    abilities are matched only by their meticulous approach to
    mathematical calculations. Whether confronting a routine calculation
    or an esoteric formula, they tackle each challenge with the same
    level of dedication and expertise.\n- Finally, with an unrelenting
    commitment to veracity, PythonProgramming_Expert rigorously verifies
    physical and mathematical results. They understand that in the
    delicate ballet of spaceflight, there is no room for error and that
    the accurate validation of results is paramount for successful
    missions. This dedication ensures that when PythonProgramming_Expert
    presents a solution, it is not only theoretically sound but also
    practically reliable."
11 },
```

We also summarized the agent-selected rate for each scenario in Figure 5. It is obvious that selected agents are highly related to the current scenarios. The verification expert has a high selection rate because we prompt Captain Agent in the system message to create a verification role to maintain the conversation. We also notice that in some specific scenarios (mathematics, data analysis, and programming), some agents with a general name and description will have a high selection rate (e.g., PythonMath_Expert, DataAnalysis_Expert, CodeReview_Expert, etc.). However, in the Science scenarios, there are no highly preferred agents with general descriptions, and the selection distribution become flatten.

# H    TOOL LIBRARY

This section provides the names and descriptions of our manually created tool library. The tools are categorized into three classes: Information Retrieval, Data Analysis and Math Problem Solving. For each category, we summarize the patterns of the corresponding dataset and manually craft a set of functions suits the tasks and can potentially enhance the agents' task resolution capability.

Table 8: Tools for Information Retrieval category.

| Tools | Description |
|---|---|
| scrape_wikipedia_tables | Scrapes Wikipedia tables based on a given URL and header keyword. |
| transcribe_audio_file | Transcribes the audio file located at the given file path. |
| youtube_download | Downloads a YouTube video and returns the download link. |
| academic_search | Perform an academic search of papers, authors or an author's papers. |
| docx_to_md | Converts a DOCX file to Markdown format. |
| pptx_to_md | Convert a PowerPoint presentation (PPTX) to Markdown format. |
| spreadsheet_to_md | Convert an Excel spreadsheet file to Markdown format. |
| extract_pdf_image | Extracts images from a PDF file and saves them to the specified output directory. |
| extract_pdf_text | Extracts text from a specified page or the entire PDF file. |
| get_youtube_caption | Retrieves the captions for a YouTube video. |
| image_qa | Answers your questions about a given image. |
| optical_character_recognition | Perform optical character recognition (OCR) on the given image. |
| perform_web_question_answering | Perform web search according to keyword and answer your question on each webpage search result, or directly on the webpage if the keyword is a URL. For each search result, a response to the question is provided. |
| scrape_wikipedia_tables | Scrapes Wikipedia tables based on a given URL and header keyword. |

Table 9: Tools for Data Analysis category.

| Tools | Description |
|---|---|
| calculate_correlation | Calculate the correlation between two columns in a CSV file. |
| calculate_skewness_and_kurtosis | Calculate the skewness and kurtosis of a specified column in a CSV file. The kurtosis is calculated using the Fisher definition. |
| detect_outlier_iqr | Detect outliers in a specified column of a CSV file using the IQR method. |
| detect_outlier_zscore | Detect outliers in a CSV file based on a specified column. The outliers are determined by calculating the z-score of the data points in the column. |
| explore_csv | Reads a CSV file and prints the column names, shape, data types, and the first few lines of data. |
| shapiro_wilk_test | Perform the Shapiro-Wilk test on a specified column of a CSV file. |

Table 10: Tools for Math Problem solving category.

| Tools | Description |
|---|---|
| calculate_circle_area_from_diameter | Calculate the area of a circle given its diameter. |
| calculate_day_of_the_week | Calculates the day of the week after a given number of days starting from a specified day. |
| calculate_fraction_sum | Calculates the sum of two fractions and returns the result as a mixed number. |
| calculate_matrix_power | Calculate the power of a given matrix. |
| calculate_reflected_point | Calculates the reflection point of a given point about the line y=x. |
| complex_numbers_product | Calculates the product of a list of complex numbers. |
| compute_currency_conversion | Compute the currency conversion of the given amount using the provided exchange rate. |
| count_distinct_permutations | Counts the number of distinct permutations of a sequence where items may be indistinguishable. |
| evaluate_expression | Evaluates a mathematical expression with support for floor function notation and power notation. |
| find_continuity_point | Find the value that ensures the continuity of a piecewise function at a given point. |
| fraction_to_mixed_numbers | Simplifies a fraction to its lowest terms and returns it as a mixed number. |
| modular_inverse_sum | Calculates the sum of modular inverses of the given expressions modulo the specified modulus. |
| simplify_mixed_numbers | Simplifies the sum of two mixed numbers and returns the result as a string in the format 'a b/c'. |
| sum_of_digit_factorials | Calculates the sum of the factorial of each digit in a number. |
| sum_of_primes_below | Calculates the sum of all prime numbers below a given threshold. |

