# OpenReview forum: "Adaptive In-conversation Team Building for Language Model Agents"
_ICLR.cc/2025/Conference — Submitted to ICLR 2025_

### Official Review · Reviewer_odMG · 2024-10-21

**Soundness:** 3
**Presentation:** 3
**Contribution:** 2
**Rating:** 5
**Confidence:** 3

**Summary:**

This paper focuses on the problem of the auto-building of LLM-based multi-agent systems. This work proposes an adaptive team-building paradigm, where a Captain Agent dynamically forms and manages a group of agents for different decomposed steps of a given task instruction. Experimental results on six datasets validate the effectiveness and cost efficiency of the proposed Captain Agent.

**Strengths:**

1. Propose a new LLM-based multi-agent framework for real-world problem-solving, which is well-motivated by the real-world team building.
2. Introduce the nested group conversation and reflection schemes among agents.
3. Experimental results on six datasets validate the effectiveness and cost-efficiency of the proposed Captain Agent.

**Weaknesses:**

1. The idea of adaptive building is kind of incremental from the idea of auto building, by just adding more "specialized" agents with tailored prompting schemes.
2. The overall framework (as shown in Fig.3) is really complicated, which may introduce high variance on the final performance due to the prompt sensitivity of LLMs. The reproducibility of the results will also be questionable.
3. The presentation of the experimental results is a bit confusing, such as the choice of datasets in different analysis, the backbone of LLMs for different frameworks, etc. (More in the questions)

**Questions:**

1. How is the respective performance on the agent retrieval and agent selection? The ablation study only shows the performance without the library.
2. Why different datasets are adopted for different analysis? For example, only the GAIA is adopted for the analysis of the agent/tool library, but the rest are used for the analysis of dynmic team-building.
3. In the cost analysis, how did you compute the cost in terms of different backbones? Is that fair? Should we focus on the cost comparison between different frameworks with the same backbones?
4. The results in other tables also contain a mix of frameworks with different backbones. It makes the results kind of confusing on whether the performance gain is from the backbone or the framework.

---

> ### Author Response · Authors · 2024-11-20
> **Author's reply to reviewer odMG**
>
> Thank you for your meaningful feedback. We answer your questions below and hope they can address your concerns.
>
> > [w1] The idea of adaptive building is kind of incremental from the idea of auto building, by just adding more "specialized" agents with tailored prompting schemes.
>
> Thank you for pointing out the novelty of our work. As highlighted in our work, previous auto-building methods can be summarized as “static build” with hand-crafted blueprints and fixed sequential processes for team building. Unlike the fixed sequential process, Captain Agent can also be involved in the nested group chat as it can solve part of the problems by itself and pass the solution into the nested chat. Furthermore, the Captain Agent can cache teams in its memory and call back at a proper time. Therefore, the Captain Agent acts like a time leaper who can participate in different teams on different timelines to help derive better solutions.
>
> > [w2] The overall framework (as shown in Fig.3) is really complicated, which may introduce high variance on the final performance due to the prompt sensitivity of LLMs. The reproducibility of the results will also be questionable.
>
> We have considered prompt sensitivity in our work in two ways: (1) prompt sensitivity when instruction changes and (2) prompt sensitivity when the model changes. For (1), we ensure all baselines receive the problem with the same instruction template, ensuring comparison fairness. Our instruction template is simple without introducing task-irrelevant information (more details in `Appendix E`). For (2), based on the results in Table 6, Captain Agent performs well with other LLM backbones like `gpt-4o-mini` (average ranking at 2.2/8) and `Llama-3-70B-Instruct` (average ranking at 4.6/8), demonstrating the robustness of our prompting design that common LLMs can accept. With the development of LLM, the reproduced results can be even better than we recorded in the paper.
>
>
> > [Q1] How is the respective performance on the agent retrieval and agent selection? The ablation study only shows the performance without the library.
>
> Thank you for your question. We perform an ablation study that removes the agent retrieval and agent selection function as the Captain Agent version of the static build in `Section 3.4.1` and record the result in `Table 4`. Our results show that Adaptive build equipped with agent retrieval and agent selection consistently outperforms the static build paradigm.
>
>
> > [Q2] Why different datasets are adopted for different analysis? For example, only the GAIA is adopted for the analysis of the agent/tool library, but the rest are used for the analysis of dynamic team-building.
>
> Thank you for pointing out the choice of dataset in ablation studies. We chose GAIA because (1) we have a limited budget to complete all experiments, and (2) solving GAIA problems highly relies on diverse agents and tools. Therefore, we choose GAIA to perform our ablation study on the agent and tool library.
>
>
> > [Q3] In the cost analysis, how did you compute the cost in terms of different backbones? Is that fair? Should we focus on the cost comparison between different frameworks with the same backbones?
>
> According to `Table 6`'s caption, all baselines are equipped with `gpt-4-0125-preview`. We compare the baselines with four versions of Captain Agent (with `gpt-4-0125-preview`, `gpt-4o-mini`, `Llama-3-70B-Instruct`, and `Llama-3-8B-Instruct`). We can conclude from the experiment results that: (1) In the same model size, Captain Agent costs more than all baselines, but (2) Captain Agent can achieve competitive performance on weak LLMs.
>
>
> > [Q4] The results in other tables also contain a mix of frameworks with different backbones. It makes the results kind of confusing on whether the performance gain is from the backbone or the framework.
>
> According to `Section 3.1 (L304)` and the caption in `Table 2` and `Table 6`, unless otherwise specified, the results in `Tables 2, 3, 4, 5, and 6` show that both Captain Agent and all baselines are equipped with the same backbone model: `gpt-4-0125-preview`. Therefore, the comparison is fair.
>
> In `Section 3.4.3`, we test Captain Agent with `gpt-4-0125-preview` and three additional models, including `gpt-4o-mini`, `Llama-3-70B-Instruct`, and `Llama-3-8B-Instruct`. Our experiments in `Table 6` show that Captain Agent equipped with `gpt-4o-mini` outperforms the baselines equipped with `gpt-4-0125-preview`, demonstrating the superiority of Captain Agent’s design.

---

> ### Author Response · Authors · 2024-11-24
> **Further Discussion**
>
> Dear Reviewer odMG:
>
> Thanks a lot for your efforts in reviewing this paper. We tried our best to address the mentioned concerns. As the discussion deadline between reviewers and authors is very close, we tend to confirm whether there are unclear explanations and descriptions here. We could further clarify them.
>
> Thanks!

---

> ### Comment · Reviewer_odMG · 2024-11-25
>
> Thanks for the reply. I am not fully convinced by the necessity of such a complicated framework. I consider that my rating reflects an appropriate assessment on this work. All the best.

---

### Official Review · Reviewer_ASLt · 2024-11-02

**Soundness:** 3
**Presentation:** 2
**Contribution:** 3
**Rating:** 8
**Confidence:** 4

**Summary:**

The authors are working in the domain of multi-agent interaction with LLMs. They mention that previous frameworks are either 1) inefficient by giving duplicate work or 2) not able to adapt as the tasks become more complex. Due to this the authors propose a multi-agent team-building paradigm called adaptive build that will not only select relevant agents and tools but also give feedback to the main builder to adjust the set of LLMs as the tasks progresses.

They name their method Captain Agent which first identifies a subtask, lists the roles needed for this subtask, puts together a team of agents (LLMs) and tools. The agents will talk to each other to try to solve the task. Once the task is complete there is a separate LLM called the reflector that provides feedback on whether to adjust the team of agents or subtask or output the results.

The authors run Captain Agent on real-world scenarios like math, data analysis, programming, science problem solving and world information retrieval. They compare their setup against other popular agent building platforms and show they outperform these frameworks in terms of accuracy.

The authors also run some ablation studies: 1) how does performance change without the agent/tool library and 2) how does performance change with different LLMs as the builder manager (gpt-4, gpt-4o, Llama etc). Additionally the authors show that their framework can be more cost effective in terms of how much it costs calling these model APIs.

**Strengths:**

1) The authors are tackling a very challenging problem and are proposing what seems to be a novel solution to this domain. The most significant contribution to me is the Reflector LLM that provides feedback to the agent builder such that it can learn a more efficient team. I also think it's great that the authors did cost analysis; however, it would be good to see what are other costs that could be measured such as latency. Is there a large increase in latency given the multitude of steps.

2) The authors do compare against a strong set of baselines. They mention some of the differences between their framework and previous ones in Appendix C which I think should be referenced in the actual text. Also it was good to see that the authors ran their framework on a diverse set of scenarios.

**Weaknesses:**

I think there can be much more clarity in the presentation of the paper and more discussion regarding certain components of the Captain Agent framework.

Regarding discussion of certain components:

1) There isn't much discussion on the impact of the Reflector LLM. I would like to know what are specific examples of errors caught by this component and quantitative results showing its impact on overall performance.

2) Also there was a mention of a memory cache, but it's unclear how it was used in the framework. I would also like to see quantitative results showing its impact on overall performance.

3) Also analysis like the number of turns in the conversation before completing the task, the number of cycles Captain Agent had to go through before completing the task.

Regarding clarity:

1) I think Figure 4 in the Appendix is great to have. It made the high-level understanding of the framework easier and should be referenced in the main text.

2) I would like to know more detail regarding how was the agent library constructed. Additionally, how much effort is required to adapt the library for a new task domain?

3) Also some notation is unclear such as what is "agent_1_3" mean.

I think if the authors clarify some of these points it would help.

**Questions:**

I listed my questions and suggestions in the Weaknesses section.

---

> ### Author Response · Authors · 2024-11-20
> **Author's reply to reviewer ASLt [part 1/2]**
>
> Thank you for your meaningful feedback. We provide additional experiment results below regarding your question about the reflector, team cache mechanism, and conversation turns. We also answer questions regarding your clarification request. We hope the experiments and answers can address your concerns and help you better understand our work.
>
> > [w1] There isn't much discussion on the impact of the Reflector LLM. I would like to know what are specific examples of errors caught by this component and quantitative results showing its impact on overall performance.
>
> Thank you for your meaningful advice. Based on the result in `Table 2`, we
> - summarize the total incorrect number for each scenario before the reflector intervenes as “# of incorrect w/o reflector, ” and
> - summarize the “# of need double-check”, which records the times the reflector detects conflict and mistakes in the conversation, and
> - summarize the  “# of correct after double-check”, which denotes the count that Captain Agent successfully solved the problem after the reflector’s intervention, and
> - calculate the “Reflector improves”, which denotes the performance gain with the reflector’s intervention.
>
> |                                 | Math   | Programming | Data Analysis | (Sci) Chemistry | (Sci) Physics |
> |---------------------------------|--------|-------------|---------------|-----------------|---------------|
> | # of incorrect w/o reflector    | 63     | 12          | 73            | 17              | 18            |
> | # of need double check          | 29     | 12          | 56            | 5               | 4             |
> | # of correct after double check | 18     | 7           | 37            | 3               | 3             |
> | Reflector success rate          | 62.07% | 58.33%      | 66.07%        | 60.00%          | 75.00%        |
> | Reflector improves              | +9.32% | +4.26%      | +14.39%       | +7.32%          | +9.38%        |
>
> Our results show that the reflector plays an important role when the conversation conflicts and provides a promising performance improvement.
>
> > [w2] Also there was a mention of a memory cache, but it's unclear how it was used in the framework. I would also like to see quantitative results showing its impact on overall performance.
>
> Thank you for your meaningful advice. Following your instruction, we summarize the number of times that Captain Agent uses the team cache and the difference in performance and cost in the following table.
>
> |                                 |  Math  | Programming | Data Analysis | (Sci) Chemistry | (Sci) Physics |
> |---------------------------------|--------|-------------|---------------|-----------------|---------------|
> | # of times build from the cache | 49     | 20          | 80            | 2               | 1             |
> | Performance w/ team memory      | 77.55% | 96.95%      | 88.32%        | 65.85%          | 53.12%        |
> | Performance w/o team memory     | 75.12% | 95.12%      | 88.32%        | 63.41%          | 53.12%        |
> | Cost change w/ team memory      | -9.68% | -7.31%      | -10.56%       | -2.9%           | -1.7%         |
>
> Our results show that cache usage rates vary from task to task. Captain Agent tends to use the cached team to solve problems when the team previously conflicts for tasks that share similar knowledge, like math, programming, and data analysis. On the other hand, Captain Agent tends to build new teams during the task-solving process for science tasks like chemistry and physics. Our team caching mechanism helps reduce the cost of sharing similar knowledge with minor changes in the performance.
>
> > [w3] Also analysis like the number of turns in the conversation before completing the task, the number of cycles Captain Agent had to go through before completing the task.
>
> Thank you for notifying us to statistic the conversation turns for Captain Agent to solve the tasks. Conversation in Captain Agent consists of two parts: (2) the conversation between Captain Agent and the user proxy and (2) the nested conversation between the built working team. We record the average turns of Captain Agent and the nested conversation when solving tasks in the following table.
>
> |                                       |  Math | Programming | Data Analysis | (Sci) Chemistry | (Sci) Physics | Avg.  |
> |---------------------------------------|-------|-------------|---------------|-----------------|---------------|-------|
> | Average turns of Captain Agent        | 3.96  | 3.60        | 3.53          | 4.19            | 4.0           | 3.86  |
> | Average  turns of nested conversation | 8.31  | 3.18        | 16.01         | 7.07            | 7.69          | 10.57 |
>
> Our results show that complex tasks like data analysis need more turns (16.01), while simple tasks like programming require fewer turns (3.18). This conclusion is consistent with our result in `Table 6`, where complex tasks require more budget.

---

> > ### Author Response · Authors · 2024-11-20
> > **Author's reply to reviewer ASLt [part 2/2]**
> >
> > > [w4] I think Figure 4 in the Appendix is great to have. It made the high-level understanding of the framework easier and should be referenced in the main text.
> >
> > Thank you for letting us know the value of `Figure 4`. We have cited the appendix section (`Appendix F`) in our main paper and clarified it in the next version.
> >
> > > [w5] I would like to know more detail regarding how was the agent library constructed. Additionally, how much effort is required to adapt the library for a new task domain?
> >
> > Thanks for raising this question. According to `Section 3.1 (L318)`, we randomly sampled about 20 problems from each task scenario as our initial set. We let Captain Agent run the generation process for each problem to initialize the agent library. This can reflect Captain Agent's generalization budget when encountering a new domain.
> >
> >
> > > [w6] Also some notation is unclear such as what is "agent_1_3" mean.
> >
> > Thanks for pointing it out. This notation is in `Figure 3`. For the role $i$ suggested by Captain Agent, a set of top-$k$ candidates will be retrieved from the agent library, termed `agent_i_1`, `agent_i_2`, …, `agent_i_topk`. At the top of `Figure 3`, `agent_1_3` is selected, which means that the third candidate is chosen for role 1.

---

> ### Comment · Reviewer_ASLt · 2024-11-20
>
> Thanks to the authors for answering my questions. I am satisfied with the answers and have adjusted my score accordingly.

---

### Official Review · Reviewer_T5qt · 2024-11-03

**Soundness:** 2
**Presentation:** 3
**Contribution:** 2
**Rating:** 5
**Confidence:** 4

**Summary:**

This paper introduces "Captain Agent," an adaptive multi-agent system designed for dynamically building and managing LLM-based agent teams to handle complex tasks. The "Adaptive Build" paradigm, driven by Captain Agent, aims to outperform static approaches by tailoring agent teams to evolving task demands. Evaluations across six real-world scenarios reportedly show significant accuracy improvements, particularly without intensive prompt engineering.

**Strengths:**

1. The adaptive team-building approach is novel and well-designed, allowing for dynamic adjustments based on task requirements.
2. The paper provides a comprehensive evaluation, demonstrating Captain Agent's effectiveness in varied tasks, from data analysis to programming.
3. The design aims to perform well with minimal prompt customization, enhancing scalability.

**Weaknesses:**

1. Performance improvements heavily rely on using GPT-4-0125-preview for Captain Agent, raising questions about whether the gains stem from model strength rather than the proposed team-building design.
2. Using GPT-4-0125-preview as the backbone for Captain Agent but not for all baselines could create an advantage that does not necessarily reflect the paradigm's effectiveness. Ensuring baselines operate with equivalent model capabilities would strengthen the fairness of the comparisons.
3. The absence of ablation studies using weaker LLM backbones for Captain Agent limits clarity on whether performance gains come from the design itself or the underlying model.

**Questions:**

As in weaknesses

**Details Of Ethics Concerns:**

N.A.

---

> ### Author Response · Authors · 2024-11-20
> **Author's reply to reviewer T5qt**
>
> Thank you for your questions and feedback. We address your concerns listed in your weakness below and hope this helps you better understand our paper.
>
> > Performance improvements heavily rely on using GPT-4-0125-preview for Captain Agent, raising questions about whether the gains stem from model strength rather than the proposed team-building design. Using GPT-4-0125-preview as the backbone for Captain Agent but not for all baselines could create an advantage that does not necessarily reflect the paradigm's effectiveness. Ensuring baselines operate with equivalent model capabilities would strengthen the fairness of the comparisons.
>
> According to `Section 3.1 (L304)` and the caption in `Table 2` and `Table 6`, unless otherwise specified, the results in `Tables 2, 3, 4, 5, and 6` show that both Captain Agent and all baselines are equipped with the same backbone model: `gpt-4-0125-preview`. Therefore, the comparison is fair, which is not contributed by the backbone model.
>
> > The absence of ablation studies using weaker LLM backbones for Captain Agent limits clarity on whether performance gains come from the design itself or the underlying model.
>
> In `Section 3.4.3`, we test three additional models, including `gpt-4o-mini`, `Llama-3-70B-Instruct`, and `Llama-3-8B-Instruct`. Our experiments in `Table 6` show that Captain Agent equipped with `gpt-4o-mini` outperforms the baselines equipped with `gpt-4-0125-preview`, further demonstrating the superiority of Captain Agent’s design.

---

> > ### Comment · Reviewer_T5qt · 2024-11-26
> >
> > Thanks for the clarifications on the backbone method. I looked through the other reviews and discussions. Still not feeling very satisfied with the work. As all methods are using gpt-4, the cost and real-world scalability issues remain there. I guess this is also part of the reasons why the largest dataset used in this work is only with 257 data samples. Considering the limited dataset size, the complexity of the proposed framework becomes more evident. It might be more beneficial if the authors can better analyze the role and quantitative impact of specific components, like the Reflector LLM and memory cache, providing more insights on guiding and inspiring further research.

---

> > > ### Author Response · Authors · 2024-11-26
> > > **Reply to reviewer T5qt**
> > >
> > > We sincerely thank you for your response. We would like to highlight that we have additional experiments of Captain Agent with different backbone LLMs in `Section 3.4.3`, where we try Captain Agent with `gpt-4o-mini`, `Llama-3-70B-Instruct`, and `Llama-3-8B-Instruct`. Our ablation results show that Captain Agent with `gpt-4o-mini` only requires less than two dollars to complete all experiments and it outperforms all baselines equipped with `gpt-4-0125-preview`. We also discuss the quantitative impact of Reflector LLM and memory cache with Reviewer ASLt, and we summarize the conclusion below:
> > >
> > > > For the impact of the Reflector
> > >
> > > Based on the result in `Table 2`, we
> > > - summarize the total incorrect number for each scenario before the reflector intervenes as “# of incorrect w/o reflector, ” and
> > > - summarize the “# of need double-check”, which records the times the reflector detects conflict and mistakes in the conversation, and
> > > - summarize the  “# of correct after double-check”, which denotes the count that Captain Agent successfully solved the problem after the reflector’s intervention, and
> > > - calculate the “Reflector improves”, which denotes the performance gain with the reflector’s intervention.
> > >
> > > |                                 | Math   | Programming | Data Analysis | (Sci) Chemistry | (Sci) Physics |
> > > |---------------------------------|--------|-------------|---------------|-----------------|---------------|
> > > | # of incorrect w/o reflector    | 63     | 12          | 73            | 17              | 18            |
> > > | # of need double check          | 29     | 12          | 56            | 5               | 4             |
> > > | # of correct after double check | 18     | 7           | 37            | 3               | 3             |
> > > | Reflector success rate          | 62.07% | 58.33%      | 66.07%        | 60.00%          | 75.00%        |
> > > | Reflector improves              | +9.32% | +4.26%      | +14.39%       | +7.32%          | +9.38%        |
> > >
> > > Our results show that the reflector plays an important role when the conversation conflicts and provides a promising performance improvement.
> > >
> > > > For the impact of memory cache
> > >
> > > Thank you for your meaningful advice. Following your instruction, we summarize the number of times that Captain Agent uses the team cache and the difference in performance and cost in the following table.
> > >
> > > |                                 |  Math  | Programming | Data Analysis | (Sci) Chemistry | (Sci) Physics |
> > > |---------------------------------|--------|-------------|---------------|-----------------|---------------|
> > > | # of times build from the cache | 49     | 20          | 80            | 2               | 1             |
> > > | Performance w/ team memory      | 77.55% | 96.95%      | 88.32%        | 65.85%          | 53.12%        |
> > > | Performance w/o team memory     | 75.12% | 95.12%      | 88.32%        | 63.41%          | 53.12%        |
> > > | Cost change w/ team memory      | -9.68% | -7.31%      | -10.56%       | -2.9%           | -1.7%         |
> > >
> > > Our results show that cache usage rates vary from task to task. Captain Agent tends to use the cached team to solve problems when the team previously conflicts for tasks that share similar knowledge, like math, programming, and data analysis. On the other hand, Captain Agent tends to build new teams during the task-solving process for science tasks like chemistry and physics. Our team caching mechanism helps reduce the cost of sharing similar knowledge with minor changes in the performance.
> > >
> > > We hope the above analysis can address your concern about `analyze the role and quantitative impact of specific components, like the Reflector LLM and memory cache`.
> > >
> > > Again, we sincerely thank all your efforts in reviewing our manuscript. We are very glad to discuss with you if you have any further concerns.

---

> ### Author Response · Authors · 2024-11-25
> **Looking forward to your feedback**
>
> Dear Reviewer T5qt,
>
> Thanks a lot for your efforts in reviewing this paper. We tried our best to address the mentioned concerns particularly regarding the comparisons between our method and the Oracle method.
>
> Could you please kindly re-evaluate our paper based on the current situation? If you have any further questions, we are also very glad to discuss them. We appreciate your careful review.
>
> Best,
>
> Authors

---

### Official Review · Reviewer_RD8k · 2024-11-04

**Soundness:** 2
**Presentation:** 3
**Contribution:** 2
**Rating:** 6
**Confidence:** 4

**Summary:**

Multi-agent systems (MAS) has shown to be superior to single agent system if constructed properly. However, designing a perfect MAS requires carefully designing the agents involved, tool integrated, communication mechanism and etc. The process can be labour-intensive and time-consuming with the outcome of a MAS comes after the execution and refine the designed system need trial-and-error as well. This paper aims to tackle this annoying issue by proposing a adaptive team building method. It introduce a captain agents dynamically forms a team and utilizes the nested group chat mechanism with a reflection agent to provide a flexible and structured approach to task-solving.

**Strengths:**

1. The paper conducts a comprehensive evaluation on their approach. Including six real-world scenarios showing that there method is superior to various baseline and they also carry out a series of ablation study on 1) static team building vs adaptive team building 2) w and w/o tool library or agent library 3) the effect of  different backbone llm 4) cost analysis.

2. The paper provides a comprehensive overview of relevant studies, effectively situating the current research within broader literature landscape.

Though the paper present certain limitations (see Weaknesses), I would rate it a 6 due to its strong empirical results. I would also be open to increasing the score if the authors address the questions and concerns raised.

**Weaknesses:**

The paper conducts an extensive set of experiments and provides a thorough analysis of its approach within the context of Multi-Agent Systems (MAS). However, the main limitation appears to be in the novelty of the proposed contributions.

1. If I understand correctly, the proposed approach heavily relies on an existing MAS framework, AutoGen, which is already designed with scalability and flexibility in mind. While this paper extends AutoGen by adding new features, these additions may lack sufficient originality to distinguish the work from the underlying framework. A clearer differentiation or unique contribution would enhance the paper's impact.

Additionally, there are some experimental details that are not fully addressed, and some claims are made without adequate supporting evidence.

1. Figure 3: It is unclear what the main distinction is between "agent retrieval" and "agent selection." If the subtask involves retrieving an agent and tool, why is this set not directly utilized to complete the subtask and need the selection process?

2. Line 230 (Cached Team): The concept of a "cached team" needs further clarification. Is the cache meant to store the entire chat history or only the configuration? And, when invoked, does the cached team resume task-solving with its prior memory?

3. Line 243 (Nested Group Conversations): The term "nested group conversation" lacks sufficient explanation. Why is it called "nested"? Does this entail initiating a sub-group conversation? Additionally, how is the order of agent's speaking determine (Do they speak sequentially or simultaneously)? Lastly, does the subtask require the captain agent to decompose it further into sub-subtasks?

4. Section 2.4 (Adaptability): Does the approach dynamically alter the agent framework or workflow during addressing specific instances? Or does "adaptive" here mean that while the team composition can be configured based on instructions, it remains fixed once problem-solving begins?

5. Line 374 (AutoGen Assistant with a fixed system message is hard to complete.): Given that the proposed method is substantially built upon AutoGen, the authors should clearly highlight the key differences. A case study illustrating the features unique to this method, and absent in AutoGen, would benefit readers.

6. Section 3.4 (Static Team Construction): Additional information is needed on how the task-specific static team is constructed.

7. Section 3.4.2 (Predefined Tool and Agent Usage): Can it be inferred that the predefined tools and agents play a significant role, and that the auto-generation of agents contributes minimally?

8. Line 534: The statement that "this new paradigm helps ensure diversity, prevents limited knowledge extraction, and reduces stereotypical outputs" requires clarification. Specifically, what do "limited knowledge extraction" and "stereotypical outputs" mean in this context? Additionally, is there supporting evidence for this claim?

**Questions:**

See Weakness.

---

> ### Author Response · Authors · 2024-11-20
> **Author's reply to reviewer RD8k [part 1/2]**
>
> Thank you for your meaningful feedback. We answer your questions below and hope they can address your concerns.
>
> > [Q1] Figure 3: It is unclear what the main distinction is between "agent retrieval" and "agent selection." If the subtask involves retrieving an agent and tool, why is this set not directly utilized to complete the subtask and need the selection process?
>
> Our agent retrieval process is based on the similarity between the Captain Agent's suggested role’s description and the agent’s description recorded in the agent library. A sentence transformer extracts the embedding (refer to `Section 3.1: Compared methods and implementation`), which we cannot ensure that the relation between role description and agent description has been correctly embedded in the sentence transformer. Imagine you are searching on Google for a solution to a specific coding error; the first result may not always be the one you want. However, the probability that the top-k (e.g., k=5) results include your desired solution is high. Therefore, we further adopt “agent selection” performed by an LLM to mimic the human decision and select the top-1 solution, i.e., the most suitable agent for the role. On the other hand, our library includes 541 agents, and each agent’s profile contains a large amount of text. Let LLM skim the whole agent library and choose the best one is not cost-efficient.
>
> > [Q2] Line 230 (Cached Team): The concept of a "cached team" needs further clarification. Is the cache meant to store the entire chat history or only the configuration? And, when invoked, does the cached team resume task-solving with its prior memory?
>
> The cached team does not record the history of previous conversations because the tasks are independent. Instead, we retain the high-level information, i.e., the agent’s profile, which includes “description,” “model,” “name,” and “system message.” “Description” is for agent retrieval and nested chat speaker selection, and the “system message” includes the agent’s persona and the general guideline for solving a problem.
>
> > [Q3] Line 243 (Nested Group Conversations): The term "nested group conversation" lacks sufficient explanation. Why is it called "nested"? Does this entail initiating a sub-group conversation? Additionally, how is the order of agent's speaking determine (Do they speak sequentially or simultaneously)? Lastly, does the subtask require the captain agent to decompose it further into sub-subtasks?
>
> We call it a “nested conversation” because the conversation between the built team members is a branch of the conversation between Caption Agent and User Proxy Agent, which is a sub-conversation. A group chat manager (an LLM) determines the speaking order. It uses team members’ descriptions and the conversation history as input to decide who should speak next. Our paper has a main experiment and two ablation experiments: the performance of (1)  only changing team members’ backbone LLM and (2) changing all roles’ backbone LLM. In the main experiment, the conversation manager will equipped with `gpt-4-0125-preview`. For ablation (1), the backbone LLM of the conversation manager is still `gpt-4-0125-preview`, while for ablation (2), it will change to other models like `gpt-4o-mini`, as we exhibit in Table 6.
>
>
> > [Q4] Section 2.4 (Adaptability): Does the approach dynamically alter the agent framework or workflow during addressing specific instances? Or does "adaptive" here mean that while the team composition can be configured based on instructions, it remains fixed once problem-solving begins?
>
> Adaptive means the Captain Agent can decide to build a team, use cache, or solve the problem itself adaptively. The team-building process is also adaptive since the Captain Agent will suggest different roles based on the problem-solving progress. Captain Agent can also be involved in the nested group chat as it can solve part of the problems by itself and pass the solution into the nested chat. Furthermore, the Captain Agent can cache teams in its memory and call back at a proper time. Therefore, the Captain Agent acts like a time leaper who can participate in different teams on different timelines to help derive better solutions.

---

> > ### Author Response · Authors · 2024-11-20
> > **Author's reply to reviewer RD8k [part 2/2]**
> >
> > > [Q5] Line 374 (AutoGen Assistant with a fixed system message is hard to complete.): Given that the proposed method is substantially built upon AutoGen, the authors should clearly highlight the key differences. A case study illustrating the features unique to this method and absent in AutoGen would benefit readers.
> >
> > AutoGen is known as a basic framework for multiple LLM agent systems. It provides the basic agent-agent and agent-user communication protocol and highly customized API for the user to develop their own agent-wise workflow, like the relationship between Pytorch and neural network structures. No one will claim that those neural networks are extensions of Pytorch by just adding new structures. Our proposed method is the same. It is based on Autogen's agent communication protocol and API to execute the tool and nested conversation functions, which is not a trivial extension of AutoGen.
> >
> > > [Q6] Section 3.4 (Static Team Construction): Additional information is needed on how the task-specific static team is constructed.
> >
> > - The “Two agents” manually build two agents and fix the agent for all tasks.
> > - AutoAgent builds a team according to the task without any change after the task-solving process starts.
> > - Meta-prompting includes a fixed agent-building process. It is a multi-agent framework that does not include the conversation part but lets the agents speak one by one and aggregate their results by a manager agent.
> > - Agentverse and DyLAN require hand-crafted blueprints for each task. They need to determine the building process for specific tasks, e.g., instructions on how to build a team to solve math problems.
> >
> >
> > > [Q7] Section 3.4.2 (Predefined Tool and Agent Usage): Can it be inferred that the predefined tools and agents play a significant role, and that the auto-generation of agents contributes minimally?
> >
> > We perform the ablation study on the GAIA benchmark, which highly relies on tool-using ability, with and without our tool library (Section 3.4.2). The results show that diverse agents and tools contribute almost the same (the contribution of diverse agents is higher than that of tools). The main reason is that the agent can create simple tools itself by using Python to improve performance on tool-related tasks, and diverse instruction can improve the agent’s ability to create task-specific tools.
> >
> > > [Q8] Line 534: The statement that "this new paradigm helps ensure diversity, prevents limited knowledge extraction, and reduces stereotypical outputs" requires clarification. Specifically, what do "limited knowledge extraction" and "stereotypical outputs" mean in this context? Additionally, is there supporting evidence for this claim?
> >
> > Thank you for your meaningful question. We will answer why our method (1) helps improve knowledge extraction and (2) mitigate stereotypical outputs.
> >
> > - Our experiments have demonstrated that LLM equipped with different personas (system messages) can improve the task-solving quality in diverse tasks, including scientific tasks. This is identical to the fact that diverse personas can help trigger diverse knowledge from LLM’s memory.
> > - On the other hand, static build will limit the diversity of an LLM’s persona, consequently limiting their problem-solving ability, as LLMs with the same persona tend to have the same stereotypes, and they will refuse to solve some specific problems or continue making the same error[1] or focusing on meaningless (or toxic) part in the instruction[2]. Therefore, improving the diversity of personas helps mitigate stereotypes in the problem-solving process.
> >
> > Refs:
> >
> > [1] Gupta, Shashank, et al. "Bias Runs Deep: Implicit Reasoning Biases in Persona-Assigned LLMs." ICLR 2024.
> >
> > [2] Wan, Yixin, et al. "Are Personalized Stochastic Parrots More Dangerous? Evaluating Persona Biases in Dialogue Systems." Findings of the Association for Computational Linguistics: EMNLP 2023. 2023.

---

> ### Author Response · Authors · 2024-11-24
> **Looking forward to your reply**
>
> Dear Reviewer RD8k:
>
> We sincerely appreciate your efforts in reviewing this paper! We tried our best to address the concerns about clarifying details. The discussion deadline is approaching now. Are there unclear explanations and descriptions here?
>
> We are highly encouraged if your concerns have been addressed.  On the contrary, if you need any more clarification, we can provide it as soon as possible before the discussion deadline.
>
> Thanks!
>
> Authors

---

### Meta-Review · Area_Chair_DH6J · 2024-12-18

**Metareview:**

The authors address the problem of how to effectively design and manage a team of LLM-based agents to solve complex tasks and propose "Captain Agent," a novel LLM agent that dynamically forms and manages teams of agents for complex tasks, using nested conversations and reflection to improve performance. The reviewers appreciate the novelty and motivation behind the idea as well as the comprehensive evaluation. They do raise several concerns, however, including the fact that the contribution is incremental, lack of clarity in the paper, and concerns regarding the experimental setup (including reliance on GPT-4, small datasets) that may lead some claims to be unsupported. The authors respond to all concerns and questions and conduct additional experiments, but the reviewers are generally not convinced.

**Additional Comments On Reviewer Discussion:**

The discussions mainly focus on clarifications on the design and relation to existing works and on further analysis of each component's impact for which the authors conduct more experiments. However, only one reviewer seems satisfied and I believe that overall this paper is not mature enough for publication.

---

### Decision · Program_Chairs · 2025-01-22

Reject